# Characterization of a Human Neuronal Culture System for the Study of Cofilin–Actin Rod Pathology

**DOI:** 10.3390/biomedicines11112942

**Published:** 2023-10-31

**Authors:** Lubna H. Tahtamouni, Sydney A. Alderfer, Thomas B. Kuhn, Laurie S. Minamide, Soham Chanda, Michael R. Ruff, James R. Bamburg

**Affiliations:** 1Department of Biology and Biotechnology, Faculty of Science, The Hashemite University, Zarqa 13133, Jordan; lubnatahtamuni@hu.edu.jo; 2Department of Biochemistry and Molecular Biology, Colorado State University, Fort Collins, CO 80523, USA; tom.kuhn@colostate.edu (T.B.K.); laurie.minamide@colostate.edu (L.S.M.); soham.chanda@colostate.edu (S.C.); 3Department of Chemical and Biological Engineering and School of Biomedical Engineering, Colorado State University, Fort Collins, CO 80523, USA; sydney.alderfer@gmail.com; 4Creative Bio-Peptides, Inc., 10319 Glen Road, Suite 100, Potomac, MD 20854, USA; mruff@creativebiopeptides.com

**Keywords:** cofilin, cofilactin rods, human iPSCs, WTC-11 cells, amyloid-β, cellular prion protein, NADPH oxidase, cytokine/chemokine receptors, synapse development

## Abstract

Cofilactin rod pathology, which can initiate synapse loss, has been extensively studied in rodent neurons, hippocampal slices, and in vivo mouse models of human neurodegenerative diseases such as Alzheimer’s disease (AD). In these systems, rod formation induced by disease-associated factors, such as soluble oligomers of Amyloid-β (Aβ) in AD, utilizes a pathway requiring cellular prion protein (PrP^C^), NADPH oxidase (NOX), and cytokine/chemokine receptors (CCR5 and/or CXCR4). However, rod pathways have not been systematically assessed in a human neuronal model. Here, we characterize glutamatergic neurons differentiated from human-induced pluripotent stem cells (iPSCs) for the formation of rods in response to activators of the PrP^C^-dependent pathway. Optimization of substratum, cell density, and use of glial-conditioned medium yielded a robust system for studying the development of Aβ-induced rods in the absence of glia, suggesting a cell-autonomous pathway. Rod induction in younger neurons requires ectopic expression of PrP^C^, but this dependency disappears by Day 55. The quantification of proteins within the rod-inducing pathway suggests that increased PrP^C^ and CXCR4 expression may be factors in the doubling of the rod response to Aβ between Days 35 and 55. FDA-approved antagonists to CXCR4 and CCR5 inhibit the rod response. Rods were predominantly observed in dendrites, although severe cytoskeletal disruptions prevented the assignment of over 40% of the rods to either an axon or dendrite. In the absence of glia, a condition in which rods are more readily observed, neurons mature and fire action potentials but do not form functional synapses. However, PSD95-containing dendritic spines associate with axonal regions of pre-synaptic vesicles containing the glutamate transporter, VGLUT1. Thus, our results identified stem cell-derived neurons as a robust model for studying cofilactin rod formation in a human cellular environment and for developing effective therapeutic strategies for the treatment of dementias arising from multiple proteinopathies with different rod initiators.

## 1. Introduction

Cofilactin rods, bundles of cofilin-saturated F-actin, are markedly elevated in the brains of human subjects with Alzheimer’s disease (AD) but not cognitively normal subjects [1,2,3]. Although synthetic amyloid-β_1–42_ oligomers induce rods in cultured rodent neurons [4], these are not nearly as potent in rod induction as Aβ dimers/trimers (Aβd/t) fractionated from the medium of a cell line expressing human amyloid precursor protein (APP) containing familial AD mutations [5,6]. Mouse models of AD based upon the expression of human APP and presenilin, each containing familial AD-associated genetic mutations, show enhanced formation of cofilactin rods [7]. Notably, a reduction in expression of either cofilin or upstream components of its dephosphorylation (activation) reduces both rod formation and cognitive deficiencies in these mouse AD models [7,8,9]. Additionally, cognitive deficits in AD mice require the expression of the cellular prion protein (PrP^C^) [10], which is also required for Aβd/t-induced rod formation in neuronal culture [11]. Furthermore, active NADPH oxidase (NOX) and a CXCR4/CCR5 receptor are required not only for Aβd/t-induced rods but also for rod formation induced by other dementia-associated factors that include the HIV envelope gp120 protein, implicated in HIV-associated neurocognitive disorder (HAND); α-synuclein, implicated in Lewy body dementia; and proinflammatory cytokines, implicated in neuroinflammation [11,12,13,14]. Notably, treatment of neurons with mixtures of these factors does not increase the ~20% of hippocampal neurons that develop rods, demonstrating that the same neuronal subpopulation is responding to each treatment [11].

To our knowledge, no human neuronal culture system has been characterized to study cofilactin rod pathology through the PrP^C^/NOX-dependent pathway, but such a system is needed for translating the efficacy and potency of rod inhibitors examined in rodent neurons to human neurons. Human embryonic stem cells (hES cells) and induced pluripotent stem cells (iPSCs) that are programmed through the expression of transcription factors to differentiate into neuronal subtypes (glutamatergic, dopaminergic, GABAergic, etc.) have been utilized for many studies that involve synaptic function [15,16,17]. A particularly convenient iPSC line (WTC-11) was developed in which a neuronal glutamatergic transcription factor gene, *NGN2*, was inserted behind a doxycycline-inducible promoter into the AAVS1 locus, and a detailed protocol has been published for its culture and differentiation [18]. Although these neurons have been characterized for the expression of mRNA transcripts and proteome analysis following stages of their morphological differentiation [19], this information for several proteins required for rod formation has not been reported. Here, we characterized and quantified the cofilactin rod response during the maturation of neurons derived from WTC-11 cells (named i^3^Neurons for integrated, inducible, and isogenic neurons [16]) and examined the expression of known components in the PrP^C^/NOX pathway. This system is applicable for testing potential therapeutics to inhibit cofilactin rod formation in human neurons.

## 2. Materials and Methods

### 2.1. Materials

All chemicals were reagent-grade and were obtained from Sigma-Aldrich (St. Louis, MO, USA) unless otherwise stated. Amyloid-β dimers/trimers were fractionated as previously described [20,21] by gel filtration on a Superdex-75 Increase 10/300 (Sigma-Aldrich) from concentrated medium of 7PA2 cells (kindly provided by Dennis Selkoe, BWH, Harvard Univ.) and utilized at a concentration of about 1 nM based upon immunoblotting compared to synthetic Aβ [6]. Dual-tropic HIV gp120_MN_ protein was from ImmunoDX (Woburn, MA, USA). AMD3100 and Maraviroc were from Santa Cruz Biotechnology (Dallas, TX, USA). NOX inhibitor TG6-227 was kindly provided by J.D. Lambeth, from Emory University. Alexa 488-phalloidin was from ThermoFisher (Waltham, MA, USA). Primary antibodies used: Protein A purified mouse monoclonal cofilin antibody (IgG, mAb22) [22], rabbit total cofilin antibody (rabbit 1439) [23], MAP2 (IgY, chicken; ab92434, Abcam, Waltham, MA, USA), NF-H (IgG, mouse; 801601, BioLegend, San Diego, CA, USA), PSD95 (IgG, goat; SC8575, Santa Cruz Biotech), VGLUT1 (IgG, rabbit; 135303, Synaptic Systems, Goettingen, Germany), 2G13 growth cone-specific antibody (IgM, mouse; NB600785, Novus Biological, Centennial, CO, USA), CXCR4 (IgG, rabbit; HIV Reagent Program ARP-11236), CCR5 (IgG, rabbit; HIV Reagent Program ARP-11232), NOX 2 (IgG, rabbit; A70477, Epigentek, Farmingdale, NY, USA), PrP^C^ for IHC (IgG, mouse; A03202, Bertin Corp., Rockville, MD, USA), PrP^C^for Western blot (IgG mouse; 189720, Cayman Chemical, Ann Arbor, MI, USA), p22 (IgG, mouse; SC130551, Santa Cruz Biotech ), p40 (IgG, rabbit; SC15342, Santa Cruz Biotech), p47/p49 (IgG, mouse; SC17845, Santa Cruz Biotech), p67 (IgG, rabbit; SC 15342, Santa Cruz Biotech ), and GAPDH (IgG, mouse; MAB374, EMD Millipore, Burlington, MA, USA). Fluorescent secondary antibodies (Alexa dyes 488 nm, 568 nm, and 647 nm excitation) raised in either goat or donkey were from ThermoFisher. Infrared secondary antibodies for Western blots were goat anti-mouse (DyLight 680; 35518, ThermoFisher) and goat anti-rabbit (DyLight 800; SA5-10036, ThermoFisher). 

### 2.2. Culture and Differentiation of Cells

#### 2.2.1. SH-SY5Y and hES Cells

The media used for SH-SY5Y and hES cell growth and differentiation are described in Appendix A.

#### 2.2.2. Human iPSC WTC-11

The protocol was modified from Wang et al. and Fernandopulle et al. [16,18]. Briefly, WTC-11 h*NGN2* cells were cultured in mTeSR Plus (Stem Cell Technologies, Cambridge, MA, USA) on 6-well plates coated with Matrigel (Corning Life Sciences, Tewksbury, MA, USA), diluted to 1x in F12-DMEM (ThermoFisher). EDTA (1 mM) in phosphate-buffered saline (PBS) was used to split undifferentiated cells. Four days before (Day −4) seeding the cells onto coverslips coated with adhesion substrates (see below), undifferentiated WTC-11 cells were triturated in 1 mM EDTA to completely dissociate them, and each well of the 6-well plate was replated into 1 well of a new Matrigel-coated 6-well plate (1:1 culture) in mTeSR Plus medium containing 10 μM Y-27632 ROCK inhibitor (STEMCELL Technologies). After overnight culturing (Day −3), the medium was replaced with fresh mTeSR Plus medium without a ROCK inhibitor. Twenty-four hours later, mTeSR Plus was replaced with N3 medium containing 2 μg/mL doxycycline. N3 medium was used for culture until cells were plated onto coverslips (Day 0). On Day 0, differentiated WTC-11 neurons (hereafter referred to as i^3^Neurons) were washed twice with 1 mM EDTA in PBS, and 1 mL Accutase (ThermoFisher) was added to each well. Cells were collected, centrifuged for 5 min at 200× *g*, and resuspended in complete homemade neurobasal medium (complete hNB) containing 5% HyClone fetal bovine serum (FBS; ThermoFisher). The homemade neurobasal medium (hNB) was made because commercial NB often has high D-serine contamination, higher than desirable L-cysteine, quite low osmolarity, and high glucose, contributing to elevated and variable spontaneous rod formation [13,24]. Thus, hNB is made with all the components of commercial NB [24] but with high purity L-serine (Bachem AG, Bubendorf, Germany), 175 μM L-cysteine, 122 mM NaCl, and 2.5 mM glucose. Complete hNB is hNB containing 1× N21-MAX (R&D Systems, Minneapolis, MN, USA), 0.5 mM GlutaMAX-1 (ThermoFisher), 25 units/mL penicillin and 25 μg/mL streptomycin from a combined reagent (ThermoFisher), and 2 μg/mL doxycycline (Sigma-Aldrich). Cells collected from all wells of the 6-well plate were combined and plated at densities between 20,000 and 200,000 neurons/well onto ethanol-cleaned German glass coverslips (12 mm diameter) coated with various adhesion substrates: vitronectin (R&D Systems); laminin (Sigma-Aldrich); poly-D-lysine (PDL) (Sigma-Aldrich); Matrigel (Corning Life Sciences); and Matrigel plus PDL in wells of a 24-well culture plate. For the next 5 days, a half-medium exchange using fresh serum-free complete hNB was performed. Starting Day 8, medium was changed every 3 days using overnight glial-conditioned complete hNB. Glial conditioning was performed by growing glia obtained from rodent pups (P0-P2) (either rats or mice) to about 80% confluency in 10 cm tissue culture plates [25,26]. The growth of glia ceased upon washing cells and incubating them overnight in complete hNB for use on human neuronal cultures. Glial cultures for medium conditioning were replaced at ~3–4-week intervals.

### 2.3. Cell Treatments

#### 2.3.1. Viral Infection

In some experiments, neurons were infected with adenoviruses expressing mRFP (Ad RedTrack; AdRT) (control infected) or wild-type mouse PrP^C^ at a multiplicity of infection (moi; virions/cell) of 30 or 100. The adenoviruses were constructed and titered as described previously [11,27] and were produced from several different viral amplifications. GFP-expressing lentivirus was made as previously described [15]. To determine the efficiency of infecting i^3^Neurons with either adenovirus or lentivirus, we used an mRFP-expressing adenovirus (AdRT) and a GFP-expressing lentivirus. Following lentivirus infection, visible fluorescent protein expression required 8–9 days, whereas expression following adenovirus infection (30 or 100 moi) took about 4 days, both being independent of the post-differentiation time of the i^3^Neurons (7 to 51 days). When quantified 10 days after infection with lentivirus (GFP) or 4 days after infection with 30 moi of adenovirus (mRFP), >99% of DAPI-stained nuclei from multiple fields were in cells expressing GFP or mRFP, respectively (Appendix A). When used to infect i^3^Neurons, adenoviruses were applied at a moi of 30 or 100, either 3 days prior to other treatments for 21-day i^3^Neurons or 4 days prior for i^3^Neurons 35 days or older. 

#### 2.3.2. Rod Inducers and Inhibitors

Antimycin (2 μM) or glutamic acid (150 μM) were added to some neuronal cultures from 100× freshly made stock solutions to induce rods through mitochondrial-dependent/energy depletion pathways. For rod induction through the PrP^C^/NOX pathway, Aβd/t, isolated from culture medium of 7PA2 cells as previously described in [5,6], was resolubilized to a 1x concentration (~1 nM) in complete hNB for addition to neuronal cultures [6]. More than 7 preparations of combined gel-filtered Aβd/t were used during this study, but within a single culture plate, the same combined batch was used. Dual-tropic HIV gp120_MN_ was prepared as a 100× solution in hNB for use in culture at 0.25 to 1 nM. AMD3100 was prepared in sterile water, and Maraviroc was prepared in absolute ethanol; they were used at final concentrations of 33 nM or 100 nM. The NOX inhibitor TG6-227 was made as a 100× stock solution in DMSO and used at 1 μM with vehicle controls [11].

#### 2.3.3. Fixation and Immunolabeling

Neurons were fixed with 4% formaldehyde (from 20%; Tousimis Research Corporation, Rockville, MD, USA)/0.1% glutaraldehyde (from 25%, Grade 1; Sigma-Aldrich) in cytoskeletal stabilizing buffer [28] for 45 min and washed 5 times with PBS. For rod immunolabeling [29], PBS was removed by aspiration and neurons were permeabilized in 100% methanol (−20 °C at time of addition) for 3 min, washed 2× with PBS and 2× with Tris-buffered saline (TBS), and then blocked for 1 h in 2–5% goat or donkey serum (chosen to match species of secondary antibodies to be used) in TBS containing 1% bovine serum albumin (BSA). When fluorescent phalloidin staining was used with cofilin immunolabeling, fixed cells were permeabilized in cold 80% methanol/20% 1× PBS for 90 sec, washed 3× with PBS, followed by 90 s with 0.05% TritonX-100 in PBS and 3 PBS washes. All antibodies were diluted in TBS containing 1% BSA. Primary antibodies were applied overnight at 4 °C. Amounts or dilutions used: mouse monoclonal mAb22 cofilin antibody (5–10 μg/mL), rabbit 1439 total cofilin antibody (1 μg/mL), MAP2 (1:1000), NF-H (1:600), PSD95 (1:200), VGLUT1 (1:600), 2G13 (1:100), CXCR4 (1:200), CCR5 (1:200), and PrP^C^ (1:150). Coverslips were washed 5× with TBS, incubated for 1 h with fluorescently conjugated secondary antibodies from appropriate species, washed 5× in TBS, and mounted on microscope slides with ProLong Diamond Antifade. DAPI, if used, was added to either the secondary antibody mixture or the mounting solution (Invitrogen, ThermoFisher). 

#### 2.3.4. Immunoblotting

Control and treated cells were lysed in lysis buffer (2% SDS, 10 mM Tris pH 7.5, 10 mM NaF, 2 mM EGTA, 10 mM dithiothreitol), and lysates were immediately heated in boiling water for 5 min and then sonicated to shear DNA. Aliquots were diluted in 4X SDS-PAGE sample buffer (0.5 M Tris–HCl pH 6.8, 2% SDS, 20% glycerol, 20% 2-mercaptoethanol, and 0.16% bromophenol blue), and proteins were resolved by electrophoresis on 10% or 15% SDS-polyacrylamide gels (mini-gel, BioRad, Hercules, CA, USA) and transferred to nitrocellulose. Membranes were blocked for 1 h (1% (*w*/*v*) BSA in TBS), followed by overnight incubation at 4 °C with primary antibodies diluted in TBS containing 1% BSA and 0.05% Tween 20 (TBST). Primary antibodies used: CXCR4 (47 and 49 kDa) (1:500), CCR5 (40 kDa) (1:500), NOX 2 (1:500), PrP^C^ (Cayman, 1:200), p22 (1:200), p40 (1:200), p47 (1:200), p67 (1:200), GAPDH (1:6000), and cofilin (rabbit 1439; 2 ng/mL). After washing and incubating blots with appropriate chemiluminescent or IR-fluor-labeled secondary antibodies, immunolabeled bands were imaged. The IR-labeled secondary antibodies were detected using a LI-COR Odyssey Infrared Imaging System (LI-COR Biosciences, Lincoln, NB, USA). CXCR4 and CCR5 immunoblots were visualized with enhanced chemiluminescence on an ImageQuant LAS500 (Cytiva Life Science. Marlborough, MA, USA). Signals were quantified with ImageJ software (Java JDK 15; NIH.gov). 

For two-dimensional immunoblotting, proteins in SDS-containing cell extracts were precipitated with chloroform/methanol [30], rehydrated in 8 M urea containing 2% IGEPAL (Sigma) and 18 mM dithiothreitol, and separated in the first dimension by isoelectric focusing on pre-cast pH 3–10 strips (Amersham Pharmacia Biotech, Piscataway, NJ, USA) according to the manufacturer’s protocol (IPGphor Isoelectric Focusing System), followed by SDS-PAGE on 15% isocratic gels. After transfer to nitrocellulose and blocking, cofilin and its phosphorylated form were visualized with affinity-purified rabbit 1439 pan ADF/cofilin antibody and IR-labeled secondary antibody by scanning with a LI-COR Odyssey. 

### 2.4. Microscopy and Image Analysis

#### 2.4.1. Imaging at 4 to 20×

Low-power (4× to 20×) images of neurons for growth and differentiation studies were obtained on an Olympus CKX53 inverted microscope with an X-CITE LED fluorescence illumination system. Some single images were obtained using a 60× NA 1.4 objective on an inverted Nikon Diaphot microscope with a CCD camera operated by Metamorph software (Molecular Devices, San Jose, CA, USA). Large arrays of stitched images and 3- or 4-channel fluorescence overlays were obtained on a BZ-X710 microscope (Keyence Corp, Elmwood Park, NJ, USA) using 20× (0.75 na) or 40× (0.95 na) Plan Apo objectives with filter cubes (Ex/Dichroic/Em) for DAPI (350/400/460); EGFP or Alexa 488 (470/495/525); mRFP or Alexa 568 (560/585/615); and iRFP/Alexa 647 or Cy5 (620/660/700) using standard resolution (2 × 2 binning). 

#### 2.4.2. Confocal Microscopy 

Spinning-disc confocal microscopy was performed on an Olympus IX83 inverted microscope with an Andor iXon camera using a 100× 1.45 na Plan Apo objective and lasers at 488, 561, and 640 nm (system integrated by Intelligent Imaging Innovations (3I; Denver, CO, USA) operating with Slidebook Software). 

#### 2.4.3. Image Analysis

Images from the Keyence microscope were captured as either single fields or image arrays (usually 7 × 7), with or without Z-stacks. If a Z-stack was used in capture, fields were converted to full-focus (projection) images, and arrays were stitched with the Keyence Image Analyzer. Confocal images, collected using Slidebook software, were exported in tif format for further analysis with Image J (NIH.gov) or Metamorph (Molecular Devices). Rod counts from images of cofilin-immunolabeled i^3^Neurons were determined manually by individuals blind to the treatment and were normalized for neurite density by dividing rod numbers by the area of either immunolabeled microtubule-associated protein 2 (MAP2, Day 35 or older) or neurofilament heavy chain (NF-H; Day 26 or younger). The same exposure times were used throughout image capture in any single experiment. Although rod formation disrupts the cytoskeleton and leads to a decline in the intensity of immunolabeling of MAP2, by thresholding the intensity of MAP2 to obtain its area within a stitched array, normalization of rods to this area was not significantly different from normalization using cell numbers (DAPI-stained nuclear area) when rods are expressed as a percentage of the maximum response within each experimental group (Appendix A). When complete arrays have no unoccupied regions of cells, the raw rod-count data as a percentage of the maximum response also show an identical pattern. However, if some region within a stitched array is devoid of cells, normalizing to the MAP2-immunolabeled area corrects rod numbers to neurite density in the cell-occupied area. Each culture plate of i^3^Neurons included treatments to give a maximum rod response (depending on the experiment, either PrP^C^, Aβd/t, or PrP^C^ + Aβd/t) as well as untreated controls, so that rod responses between plates in a single experiment and with identical treatments across multiple experiments could be combined based upon the percentage of the maximum response. 

Confocal images of immunolabeled pre-synaptic and post-synaptic markers (VGLUT1 and PSD95, respectively) were captured using Slidebook 2023-1 software (Intelligent Imaging Innovations). Images were deconvolved and exported as tif files for manual analysis of contacting or overlapping pre- and post-synaptic puncta using Metamorph software. The images were also analyzed by spatial cross-correlation, as described in Appendix A.

#### 2.4.4. Electrophysiology

Whole-cell patch-clamp recordings of stem cell-derived human neurons were performed using an integrated patch-clamp amplifier (Sutter Instrument, Novato, CA, USA) with a customized Igor Pro (WaveMetrics, Portland, OR, USA) data acquisition system, similarly to that described before [31]. In brief, the cells were patched in using an internal solution containing 130 mM KCl, 10 mM NaCl, 2 mM MgCl_2_, 0.5 mM EGTA, 0.16 mM CaCl_2_, 4 mM Na_2_ATP, 0.4 mM NaGTP, 14 mM Tris-creatine phosphate, and 10 mM HEPES (pH 7.3, adjusted with KOH; osmolarity ≈ 310 mOsm). The extracellular bath solution contained 140 mM NaCl, 5 mM KCl, 3 mM CaCl_2_, 1 mM MgCl_2_, 10 mM glucose, and 10 mM HEPES (pH 7.4, adjusted with NaOH; osmolarity ≈ 300 mOsm). Recordings for action potential (AP) firing were performed in current-clamp mode while maintaining the cells at −60 mV by continuously injecting small holding currents to adjust their membrane potentials (V_m_). The cells with AP were then switched to voltage-clamp mode, and recordings for voltage-gated Na^+^/K^+^ channels were conducted at V_m_ = −70 mV to 50 mV, with an increment of 5 mV step-pulses.

### 2.5. Statistics

Rod counts were normalized to NF-H immunolabeling (Day 26 or younger) or MAP2 immunolabeling (Day 35 or older) from independent cultures from multiple experimental groups, each having within them untreated and/or AdRT adenovirus-treated controls, Aβd/t-treated cultures, and/or adenoviral-expressed PrP^C^ ± Aβd/t. Major experiments using two hundred or more independent i^3^Neuron cultures taken for 55 days were performed four times but did not include every treatment. Since up to a dozen 24-well culture plates were used within every experimental repeat, there were several batches of Aβd/t (over 50 separate column chromatography runs, 10 of which were combined to yield 100 mL of 1× Aβd/t for a single experiment); multiple changes of glia (new cultures set up every 3 weeks) for making conditioned medium; and multiple batches of homemade NB medium made 2 L at a time. We kept batches of reagents identical for each 24-well plate but not across all plates in any one experiment. One-way ANOVA was used for significance across multiple samples within each 24-well culture plate. For combining multiple experimental groups, each coverslip within a plate was expressed as a percentage of the average of the maximum responding cultures (always either ectopic PrP^C^ ± Aβd/t or Aβd/t alone) set at 100%. The normalized results from each independent culture were entered into Prism GraphPad (Dotmatics, Bishop’s Stortford, UK) for statistical analysis using Student’s paired two-tailed t-test between two treatment groups to obtain values of significance, with a *p* < 0.05 being taken as statistically significant. Each data point on the bar graphs represents an independent culture. Unless stated otherwise in the figure legend, data points within each bar come from three or more experiments, with 1 to 4 cultures from each experiment. The same labeling was used for significant differences between groups: * *p* < 0.05; ** *p* < 0.01, *** *p* < 0.001, **** *p* < 0.0001. 

## 3. Results

### 3.1. Formation of Cofilactin Rods in Human Neurons

SH-SY5Y cells: Post-mitotic and neuronally differentiated SH-SY5Y cells (see Appendix A) develop long neurites that immunolabel for MAP2 and/or Tau. These cells form cofilactin rods in response to energy depletion but not in response to stimulators of the PrP^C^/NOX-dependent pathway such as Aβd/t or HIV-derived dual-tropic gp120_MN_ [11,13]. Although SH-SY5Y cells express an active, ROS-producing NOX complex [31], they do not express physiologically relevant levels of PrP^C^ [32]. Given their diverse and imprecisely regulated electrophysiological properties that make them unsuitable for studies of synapse formation and function [33,34], they are not a useful model for studies on the long-term effects of rods. 

Glutamatergic neurons: Neurons derived from human embryonic stem cells (hES cells) following lentiviral-mediated expression of Ngn2 [15] generated rods when treated with excitotoxic levels of glutamate (Appendix A), similar to the tandem arrays observed in glutamate-treated rodent hippocampal neurons [1]. However, treatment with Aβd/t failed to induce a rod response in these neurons cultured for up to 28 days. To further explore the use of human neurons from a stem cell origin, we selected the human iPSC-derived WTC-11 line, into which a doxycycline-inducible, isogenic, integrated *NGN2* [16,18] was inserted, thus eliminating the necessity of using the lentiviral infection for neuronal transformation. Doxycycline-induced neurons from the WTC-11 line are referred to as i^3^Neurons [16].

### 3.2. Optimizing Substrate and Cell Numbers for Differentiation of Human WTC-11 Cells

Matrigel was previously identified as the best substrate for the growth of undifferentiated WTC-11 cells on plastic [16,18,19]. However, to develop optimal conditions for the growth of i^3^Neurons from WTC-11 cells, we evaluated, both independently and combined, the substratum, cell density, and the requirement for glial cell co-cultures versus glial-conditioned medium for i^3^Neuron growth on glass coverslips using both morphology and rod formation as end points. One goal in so doing was to develop a glial-free culture system in which intraneurite rods could be readily observed by cofilin immunolabeling without interference from immunolabeling of non-neuronal cells.

Initially, on Day -2 (see Appendix A for timeline), WTC-11 cells from all wells of a 6-well plate were collected and combined with one vial (~10^6^ cells) of P0 mouse glia and plated on Matrigel-coated 12 mm coverslips distributed in all wells of a 24-well culture plate. Three days later, the glia monolayer started “shrinking” and the i^3^Neurons showed signs of stress; by seven days, most i^3^Neurons were fragmented and dead. For this reason, we tested different densities of i^3^Neurons (from 20,000 to 200,000 per 2.01 cm^2^ well of a 24-well culture dish) plated onto 12 mm diameter glass coverslips coated with different adhesion substrates, including vitronectin, laminin, PDL, Matrigel, and Matrigel plus PDL. Regardless of the adhesion substrate, densities of i^3^Neurons of less than 40,000/well resulted in cell death with accumulation of cell debris, whereas a density of ≥100,000/well resulted in large aggregates of cells with bundles of fasciculated neurites that further enlarge during 2–3 weeks of culture. Plating densities of 60,000 to 80,000 i^3^Neurons per well gave well-distributed cells with healthy outgrowth from many individual neurons but also with small (5–20 neuron) aggregates. Additionally, glia monolayers grown on laminin, Matrigel, PDL, or vitronectin started to shrink within several days, forming empty patches within the monolayer. However, cultures on Matrigel plus PDL maintained the glia monolayer with well-isolated neurons and neurites (Figure 1). The same substrate worked well for i^3^Neurons grown without glia. However, i^3^Neurons without glia started deteriorating (neurite beading and fragmentation) after ~10 days. The deterioration was prevented by feeding the i^3^Neurons, starting on Day 8, with complete hNB medium that was conditioned overnight on an 80% confluent rodent glial monolayer. 

### 3.3. Cofilactin Bundles in Filopodia of Human i^3^Neuronal Growth Cones

Scoring rods in rodent neuronal cultures is often complicated due to the presence of an immunolabeled cofilin background from both non-neuronal cells and cofilactin filament bundles in proximal regions of growth cone filopodia, where they play an important role in filopodial function [29,35,36]. It was not surprising to find that growth cones of i^3^Neurons have identical cofilactin bundles in proximal filopodia in which fluorescent phalloidin stains out to the distal tip (Figure 2A). We examined Day 10 and Day 55 i^3^Neurons immunolabeled for cofilin, NF-H, and a growth cone-specific antibody, 2G13 [37] (Figure 2B). Growth cones often migrate along other neurites, and when they collapse, as occurs in cycles during their growth, the cofilin-immunolabeled growth cone becomes difficult to distinguish from an intraneurite rod, as can be observed in Figure 2B, growth cone 4, especially when viewed at the lower magnification used for scoring rods. Thus, we exclude from scoring “rod-like” cofilin immunostaining that is at the end of a neurite immunolabeled with either NF-H or MAP2. Although small puncta of 2G13 immunolabeling such as those seen in Figure 2B are found in images of i^3^Neurons through Day 55, structures with growth cone morphology have almost entirely disappeared by the time i^3^Neurons develop a strong rod response to Aβd/t, simplifying the scoring of rods in these cultures compared with those of Day 5–7 rodent hippocampal neurons, which are commonly used for rod assays. 

### 3.4. i^3^Neuronal Development 

#### 3.4.1. Neurite Differentiation

i^3^Neuron outgrowth and neurite specification as axons and dendrites were followed for several weeks in glial-conditioned medium. The cultures were immunolabeled for MAP2 and NF-H, mature dendritic and axonal markers, respectively [38,39]. The neurites of immature neurons immunolabel for both MAP2 and NF-H. i^3^Neurons start to show neurite specification between 2 and 3 weeks in culture but increase rapidly by Day 35, when most neurites could be distinguished as either axons or dendrites based on their immunolabeling (Figure 3A). Furthermore, dendritic spines, post-synaptic structures of glutamatergic neurons, were evident along many dendrites in Day 50–55 cultures when observed at high magnification by confocal microscopy (Figure 3B, MAP2; arrows). Dendrites and axons often run in parallel (Figure 3B, MAP2/NF-H; arrows). 

#### 3.4.2. Neuronal Survival

The survival of plated neurons was determined at Day 55 from the density of cells plated (60,000 per well, or 299 cells/mm^2^) and the average number of DAPI-labeled nuclei counted in 7 × 7 stitched arrays (12.3 mm^2^) from 10 control wells in plates from three different experiments. The cell number per array should be 3678 for 100% survival, and we determined an average of 3280 ± 475 (sd) or, on average, about 90% survival.

### 3.5. Cofilactin Rod Formation

Development of a rod response. Rod induction was followed over time in cultures of i^3^Neurons. Rods were readily induced by antimycin A or excitotoxic levels of glutamate in i^3^Neurons at Days 9 and 17 (Day 17 treated with antimycin is shown in Figure 4). Even at Day 7, the neurites of i^3^Neurons are dense and not easily traced back to their soma (Figure 3A, Day 7). Cell numbers are relatively few per field, but some small clusters of soma are present by Day 7 (shown by nuclear DAPI staining, Figure 4). Rods are easily discernible, but only in magnified views (Figure 4B). Based on rod and nuclei counts from antimycin-treated Day 17 i^3^Neurons, it is reasonable to conclude that the majority of i^3^Neurons form rods in response to energy depletion, like the >80% containing rods reported for rodent hippocampal neurons grown at low density in which individual neurons remain well separated [1]. 

The i^3^Neurons cultured for different time periods were next evaluated for their spontaneous rod formation (untreated controls) and their ability to form rods in response to treatment with Aβd/t (Figure 5). 

Some spontaneous rods were found in the cultures by Day 10, but rod numbers did not increase in Day 17 or younger treated for 24 h with Aβd/t (Figure 6). Previous work has shown ectopic expression of PrP^C^-induced rod formation in rodent neurons [11]; thus, we utilized AdPrP^C^ and a control adenovirus (AdRT) to infect the i^3^Neurons to determine if they exhibited a prion-dependent rod response. Over 99% of i^3^Neurons treated with a moi of either 30 or 100 of AdRT became infected and expressed the fluorescent protein (determined by complete overlap of mRFP expression with DAPI-stained nuclei; Appendix A). A plateau in expression measured using mRFP fluorescence was reached by 4 days post-infection, the minimum post-infection time used for experiments involving adenoviral-infected neurons. The infection of i^3^Neurons with the control adenovirus did not induce rods over uninfected cultures (Figure 6).

The addition of Aβd/t to either control-infected or AdPrP^C^-infected i^3^Neurons cultured for 10 to 17 days did not significantly increase rod response (Figure 6). i^3^Neurons expressing PrP^C^ developed a weak but significant rod response that was about 2.5-fold greater than that of the controls but was not significantly increased when combined with Aβd/t. Day 21–26 i^3^Neurons without or with ectopic PrP^C^ developed a rod response to Aβd/t that increased with culture age (Figure 5 and Figure 6). By Day 55, the rod response of i^3^Neurons to Aβd/t was completely independent of ectopic PrP^C^ (Figure 6). The ratios of the rod response between treatments at various ages (Figure 6) clearly show that the response to the ectopic expression of PrP^C^ (red over blue) declines from 2.7 to 1.7. However, the rod responses of Aβd/t over the control (green over blue) and Aβd/t over ectopic PrP^C^ (green over red) increase about five-fold each over 55 days. The decline of the ratio of Aβd/t + PrP^C^ to PrP^C^ (red-green hatched over red) from about 2.5 to 1 demonstrates the age-dependent development of an independent Aβd/t response. 

In cultures of dissociated rodent hippocampal neurons, about 20–25% of neurons form Aβd/t-induced rods [6,11]. Although these are primarily glutamatergic excitatory neurons, they are mixtures of neuronal subtypes. To determine if mature glutamatergic i^3^Neurons have a more uniform rod response, we quantified the number of rods per nucleus in several stitched arrays following a 24 h treatment with Aβd/t. Per 7 × 7 array, raw rod counts seldom were over 800, whereas nuclei averaged 3280, suggesting that Aβd/t induced a rod response in a maximum of 25% of i^3^Neurons. However, tandem rod arrays (two to five rods) often occur, making the actual percentage of i^3^Neurons that form Aβd/t-induced rods much lower. Nevertheless, ~2000 rods were quantified in some fields of Day 55 i^3^Neurons treated for 24 h with Aβd/t (with or without ectopic PrP^C^) compared to <150 (7% of max.) in the corresponding parallel untreated cultures of similar density (Figure 7).

### 3.6. Developmental Changes in Expression of Rod Response Proteins in i^3^Neurons 

**Cofilin:** Cofilactin rod formation depends upon a pool of active dephosphorylated (ser3) cofilin [1,40]. Total cofilin normalized to GAPDH on immunoblots undergoes a decline (~30%) in i^3^Neurons over 51 days in culture, but when compared to the ratio at Day 21, when the Aβd/t response first becomes significant (*p* < 0.05), none of the total cofilin levels in cultures younger or older than Day 21 differ significantly (Figure 8A). The relative amounts of active and inactive (phosphorylated Ser3) cofilin during the first 3 weeks of culture were determined from 2D immunoblots. Notably, the active pool of cofilin increases from 20% to 80% of the total by Day 10 (Figure 8B) and, when analyzed within extracts from Day 21 cultures, does not change significantly even when i^3^Neurons are (1) treated for 24 h with Aβd/t; (2) after 4 days of expression of PrP^C^; or (3) both treatments combined. Although some of the values shown in Figure 8B were obtained from single 2D immunoblots, a few treatments utilized triplicate blots to demonstrate the reproducibility achieved when a single pan-ADF/cofilin antibody is used to quantify both of the separated cofilin species on one blot [23,41]. Taken together, the large pool of active cofilin and the ability of energy depletion or glutamate to generate a strong rod response suggest that the inability of Aβd/t to induce rods in i^3^Neurons before 3 weeks of culture does not arise from a deficiency in the total amounts or activity of cofilin. 

To determine if other known components of the rod signaling pathway undergo a change in expression that correlates with the development of a rod response to Aβd/t, we analyzed their expression levels using 1-dimensional immunoblotting normalized to GAPDH as the internal loading control (Figure 9). 

**PrP^C^:** PrP^C^ declined 30–50% within 10 days of i^3^Neuron differentiation but then remained relatively unchanged up to Day 35, by which time a response to Aβd/t had already occurred. Although ectopic PrP^C^ by itself induces rods, rod induction during maturation of i^3^Neurons expressing ectopic PrP^C^ declines from 2.7- to 1.7-fold over spontaneous rods (controls) (Figure 6B). PrP^C^ may be limiting during the early development of the rod response, but the significant doubling in its expression between Days 35 and 51 may contribute to the independence of Aβd/t-induced rods from ectopic PrP^C^ expression. 

**NADPH oxidase (NOX)**: Normalized levels of NOX 2 and several of its essential subunits, including p22^PHOX^, p47^PHOX^, and p67^PHOX^, do not undergo changes in expression that would suggest that they are a limiting factor for Aβd/t-induced rod formation. 

**Cytokine/Chemokine Receptors CXCR4 and CCR5**: The levels of CCR5 remained relatively constant over 51 days. CXCR4 levels were constant until Day 35. However, the significant doubling in CXCR4 between Day 35 and Day 51 might be a contributing factor for the doubling of the rod response over this period (Figure 6B). 

### 3.7. Effects of CXCR4, CCR5, and NOX Inhibitors on Rod Formation 

Inhibiting NOX activity in rodent neurons with TG6-227 blocks rod formation by any activators of the PrP^C^/NOX pathway [11]. Rod formation in rodent neurons exposed to monotropic forms of the HIV gp120 protein that signal through either CXC4 or CCR5 is inhibited by their respective receptor antagonists, AMD3100 for CXCR4 [42,43] or Maraviroc for CCR5 [44], but not by the other, whereas rods induced by the dual-tropic gp120_MN_ protein were significantly inhibited by AMD3100 alone [13]. In Day 55 i^3^Neurons, the rod response to 24 h treatments with the antagonists alone either was not or was only slightly (*p* = 0.048) different from the untreated control cultures (Figure 10). i^3^Neurons were treated for 5 days (Days 50–55) with Aβd/t, and in its continued presence, AMD3100 or TG6-227 was added for the last 24 h. Both antagonists reduced the rod response to control levels (that of the antagonist alone), indicating a reversal of rods in the presence of Aβd/t (Figure 10). Maraviroc was tested for rod inhibition when added with Aβd/t for 24 h, and it prevented Aβd/t-induced rod response. 

### 3.8. Localization of Aβd/t-Induced Rods to Neurites

Confocal microscopy at 100× was used to determine if Aβd/t rods were induced in dendrites (MAP2-positive) and/or axons (NF-H-positive) in Day 55 i^3^Neurons (Figure 11). 

Because rod formation disrupts cytoskeletal structures [1], the positive identification of the neurite containing the rod is often difficult. From multiple coverslips of several different experiments, 216 fields were examined to determine the location of 596 rods. Of all rods examined, 333 (56%) were identified as being within a dendrite based upon MAP2 immunolabeling (often very weak) on either side of the rod with no apparent NF-H label (Figure 11A). Conversely, only two rods were identified as axonal, in which NF-H immunolabel appears to be on either side of a rod without MAP2 immunolabel (Figure 11B), though in these cases, there are immunolabeled MAP2 processes next to the rod. An enlarged region of one of these rods suggests it is in an axon (dark blue) wrapping around a dendrite. However, the neurite localization of 248 rods (42%) is ambiguous because the fasciculation of axons and dendrites leads to faint immunolabeling of both MAP2 and NF-H at either end of the rod. Furthermore, 13 rods (2%) had no MAP2 or NF-H immunolabel nearby (Figure 11C), further supporting the major cytoskeletal disruption that occurs within neurites in which cofilactin rods form. 

### 3.9. Maturation of i^3^Neurons without Astrocytes: Development of Synaptic Specification and Functional Neuronal Properties

Most of our experiments with i^3^Neurons were performed in glia-free cultures, mainly to avoid any background cofilin immunolabeling in astrocytes. Although the i^3^Neurons co-cultured with glia acquire functional maturation accompanied by synapse formation and electrical activities [18,45], it remains unclear whether these cells can also develop neuronal properties even without glia, especially during the prolonged culture period in our assays. To address this question, we performed patch-clamp recordings from glia-free i^3^Neurons (Figure 12). We found that i^3^Neurons successfully matured to generate repetitive action potentials (APs) as measured in current-clamp mode (Figure 12A) and exhibited robust voltage-gated Na^+^ K^+^ channel currents in voltage-clamp mode (Figure 12B). Despite the absence of glia, the dendritic processes of the i^3^Neurons (Figure 12C) developed prominent post-synaptic structures containing PSD95, of which about 19.2 ± 5.2% overlapped with pre-synaptic VGLUT-containing puncta (Figure 12D,E; Appendix A), suggesting bona fide synaptic organization begins in the complete absence of astrocytes. Given that all PSD95 puncta are along MAP2-labeled dendrites, and these occupy only 14.6 ± 2.9% (Appendix A) of the total field, the percentage of PSD95/VGLUT contacts far exceeds those that would arise from random interactions based on the VGLUT area as a percentage of the total (1.78 ± 0.39%). We also applied a radial-distance cross-correlation analysis to demonstrate that the association of the puncta of VGLUT and PSD95 is non-random (Appendix A). These results suggest that differentiated i^3^Neurons can retain their cellular fate and identity irrespective of glial co-culture, which justifies their use in subsequent rod assays.

## 4. Discussion

Cofilactin rod formation is a general response of most vertebrate [46] and invertebrate [47] cells to energy depletion, perhaps to preserve ATP by reducing actin dynamics, a heavy consumer of ATP [3]. This mitochondrial-dependent rod-inducing pathway can also be initiated in anoxic rodent hippocampal brain slices [48] and in glutamatergic neurons by treatment with excitotoxic levels of glutamate (Appendix A). It occurs during ischemic brain injury [49,50], affects most neurons within the ischemic region, and is independent of the PrP^C^/NOX pathway. Although activators of the PrP^C^/NOX-dependent pathway generate rods in less than 25% of rodent hippocampal neurons, these appear to be directly linked to the hippocampal-dependent cognitive dysfunction occurring in rodent models of many human neurodegenerative diseases [3]. 

Rod formation is accompanied by an increase in the pool of active (dephosphorylated) cofilin and can arise globally—such as during energy depletion [1] or in response to cofilin overexpression [51] or expression of the phospho-cofilin phosphatases slingshot-1L or chronophin [6]—as well as locally, such as in response to PrP^C^-dependent rods [4,11]. However, we show here that a significant shift in the active cofilin pool, from 20% in the freshly plated (Day 0) i^3^Neurons to 80% by Day 10 post-differentiation, is insufficient by itself to drive rod formation even in the presence of the PrP^C^/NOX pathway activator Aβd/t. The immunoblot results suggest that no single protein identified so far in the PrP^C^/NOX rod-inducing pathway appears to be the sole limiting factor in the development of the rod response to Aβd/t. What other factors might play a role?

The formation of rods in vitro has been achieved by molecular crowding in mixtures of actin and ADF or cofilin, using either the hanging drop method associated with protein crystallization [40] or the addition of methylcellulose [52]. In the latter study, mixtures of cofilin and actin alone led to long, curved bundles of cofilactin filaments, reminiscent of the cofilactin bundles refractive to phalloidin staining that have been found underlying the nuclear envelope of cultured mammalian U2OS cells expressing EGFP-β-actin or EGFP-cofilin [53]. An analysis of rods in fixed cells using electron microscopy tomography suggested that they contain a distribution of filaments with a median length of about 200 nm, or approximately 80 actin monomers [40]. The addition of actin-interacting protein 1 (Aip1, also known as WDR1), a protein that severs cofilactin filaments, to the in vitro mixtures of cofilin and actin used in molecular crowding studies led to an Aip1 concentration-dependent shortening of the curved bundles to form linear and shorter rods with tapered ends, somewhat similar morphologically to those in neurites [52]. Thus, we hypothesize that signaling via the PrP^C^/NOX pathway affects local solvation–depletion forces that cause molecular crowding, driving rod formation as well as activating/recruiting severing proteins such as Aip1. A simple overexpression of PrP^C^ alone appears to drive this process in human neurons, albeit somewhat weakly, well before externally applied rod inducers can mediate rod formation. 

Because rods induced by both proinflammatory cytokines (e.g., TNFα) [11] and Aβd/t (Figure 10) undergo reversal upon inhibition of NOX as well as by the CCR antagonists, it seems likely that rod signaling is a stochastic process in which PrP^C^-containing membrane domains organize a signaling complex that contains multiple interacting components that work together. The non-additive rod response from multiple rod inducers suggests that only one subpopulation of neurons has enough of some limiting factor(s) to generate the signaling platform or achieve the magnitude of response that is necessary to form rods [11]. Also, it is not clear if every PrP^C^/NOX complex will be limited by the same component(s). Whatever the factors might be, i^3^Neurons make an excellent model in which to pursue these studies since the iPSC WTC-11 cells are amenable to genetic manipulation prior to their post-mitotic neuronal conversion [19,54]. 

The finding that >99% of all rods for which we could establish location in Day 55 i^3^Neurons reside within dendrites is also significant and might point to a reason why loss of circuitry is a slowly developing process. A single rod within an axon would likely remove that neuron completely from its local network by restricting vesicular trafficking [4,51,55], whereas in a dendrite, synapses on other dendrites or even within the rod-containing dendrite but proximal to the rod might still function provided that adequate levels of cofilin remain to modulate the plasticity of dendritic spines [56], which are functionally damaged by Aβd/t [21]. Actin dynamics drive trafficking of post-synaptic glutamate receptors as well as alterations in the size of dendritic spines that accompany long-term potentiation (LTP), a model for learning and memory [57,58,59]. However, an increase in spine size can occur independently from the ion channel insertion that leads to LTP [60]. 

i^3^Neurons are an ideal model in which to study the function of rods in human cells. These neurons can be grown without glia, making many types of microscopic observations, such as rod scoring after immunolabeling, easier than when performed in glia-containing cultures that have a high cofilin immunolabel background. Rod effects on synaptic development can be studied at both the morphological and electrophysiological levels using glial co-cultures if a rod reporter [55] is expressed from a neuronal-specific promoter, as shown in Appendix A for synapsin promoter-driven cofilinR21Q-mRFP in neurons derived from hES cells. A useful photoactivatable inducer for rod formation, CofActor, has been developed [61,62] that could be utilized in i^3^Neurons to follow synaptic development and function distal to an induced rod. The original development of the WTC-11 iPSC model was for its screening potential for compounds that would lower tau pathology [16]. Here, we demonstrate that this system is also useful for studying therapeutics focused on reducing cofilactin rod pathology, a common feature in AD [1,2], and in mouse models of AD [7,8] and other human cognitive disorders [14,63,64]. Furthermore, we have recently applied this model system to study potential novel therapeutics that block and reverse cofilactin rod formation [65].

## Figures and Tables

**Figure 1 biomedicines-11-02942-f001:**
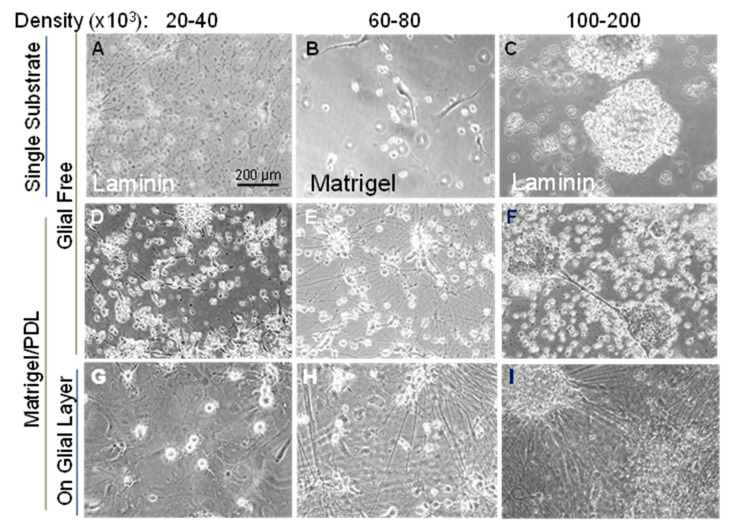
Effects of substrate and cell density on Day 7 i^3^Neurons. Day 7 i^3^Neurons plated at densities shown above in the absence (**A**–**F**) or presence (**G**–**I**) of glia. Phase-contrast images taken with a 10× objective. Top row: i^3^Neurons grown on single substrate (laminin or Matrigel). Cell debris and poor survival occur in cultures at lower densities on all substrates tested, and large aggregates of cell bodies occur at cell densities over 100 K. The combination of Matrigel/PDL (**D**–**I**) worked best for i^3^Neurons grown without (**D**–**F**) or with (**G**–**I**) glia co-culturing. Initial outgrowth is more robust on glial monolayers, but if glial-conditioned complete hNB medium is utilized starting on Day 8, healthy neurons with robust outgrowth are maintained.

**Figure 2 biomedicines-11-02942-f002:**
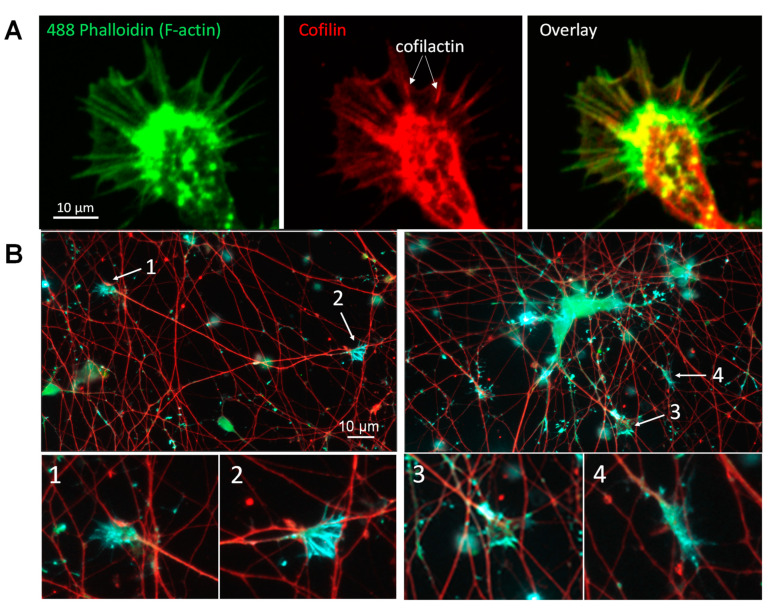
Growth cones in i^3^Neurons contain cofilactin bundles. (**A**) As in rodent neurons, cofilactin bundles in well-spread human growth cones localize to proximal regions of filopodia (arrows), whereas fluorescent phalloidin staining of F-actin (green) is brightest within the growth cone interior and extends from the cofilactin region to the distal filopodia but is weakest within the cofilactin regions (arrows in middle panel), which are brighter red in the image overlay. Images are from a single plane of a confocal image stack taken with a 100× objective. (**B**) i^3^Neurons at Day 10 contain many growth cones that immunolabel for cofilin (green) and the 2G13 antibody (pseudocolored turquoise). NF-H immunolabel is in red. Most growth cones grow along other neurites, as shown in the top two panels. Growth cones 2 and 4, shown enlarged in the lower images, could easily be mistaken for cofilactin rods. Thus, rod-like cofilin staining occurring at the end of a neurite is not included in rod quantification. Images: 20× objective, Keyence microscope.

**Figure 3 biomedicines-11-02942-f003:**
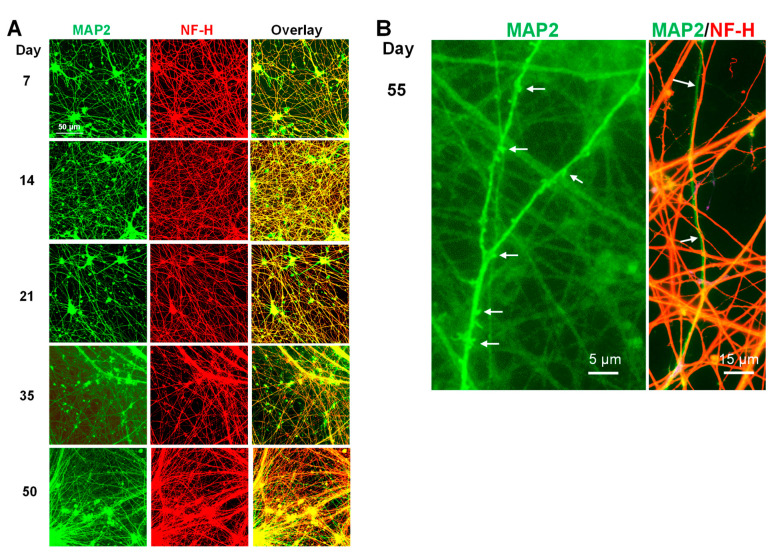
Developmental changes in i^3^Neurons cultured for 55 days. (**A**) Glial-free i^3^Neurons from Days 7–50 immunolabeled for MAP2 and NF-H. A few neurites labeling for MAP2 but not NF-H are evident by Day 21, but by Day 35, most neurites can be identified as axons (NF-H-positive, MAP2-negative) or dendrites (MAP2-positive, NF-H-negative). Images are from a Keyence microscope (20x objective). Scale bar (50 µm) applies to each image. (**B**) Left image of MAP2-immunolabeled Day 55 i^3^Neurons (100× objective on confocal microscope) shows dendritic spines (arrows). In the right image, an axon (NF-H-positive, red) and dendrite (MAP2-positive, green) run side by side (arrows).

**Figure 4 biomedicines-11-02942-f004:**
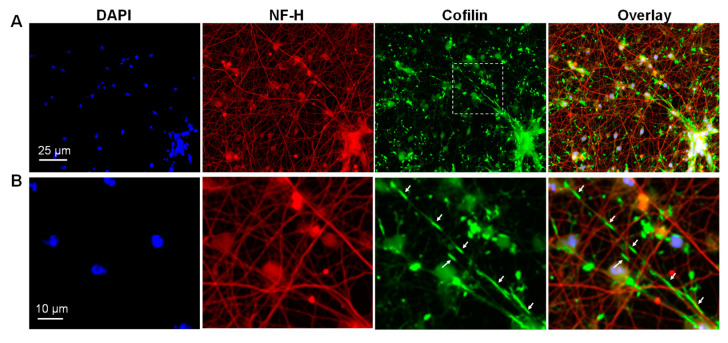
Energy depletion induces rods in i^3^Neurons. (**A**). Day 17 i^3^Neurons treated with antimycin A for 2 h. Many cofilin-stained rods are visible, often in tandem arrays. (**B**). Enlarged boxed region from the cofilin panel (top row) with color-separated images. The immunolabeling of NF-H becomes very faint in the regions of neurites in which cofilin-immunolabeled rods (white arrows) form. Images acquired with a 40× air objective.

**Figure 5 biomedicines-11-02942-f005:**
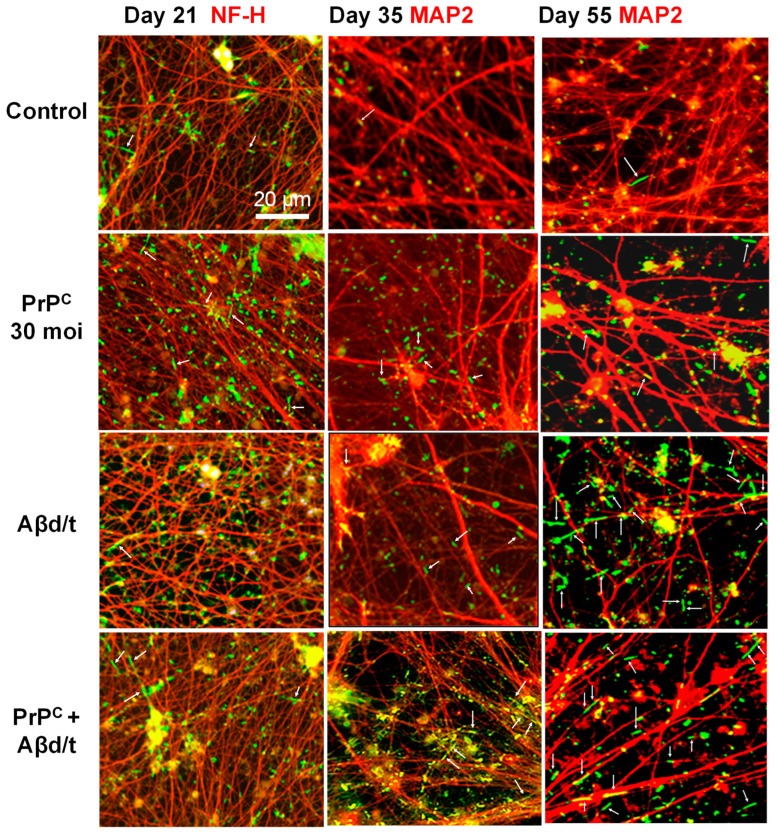
Qualitative observations of rods induced by ectopic PrP^C^ expression, Aβd/t, or their combined treatment, compared to untreated (control) i^3^Neurons at different ages. Images are enlargements of same-size regions from stitched arrays obtained on a Keyence microscope with a 20× objective; immunolabeled cofilin (green with rods identified using white arrows) is shown with either immunolabeled NF-H (Day 21) or MAP2 (Days 35 and 55) (red). A few spontaneous rods occur in all cultures, but rod numbers increase with induction, especially as i^3^Neurons mature. Rod quantification under these same conditions is shown in Figure 6.

**Figure 6 biomedicines-11-02942-f006:**
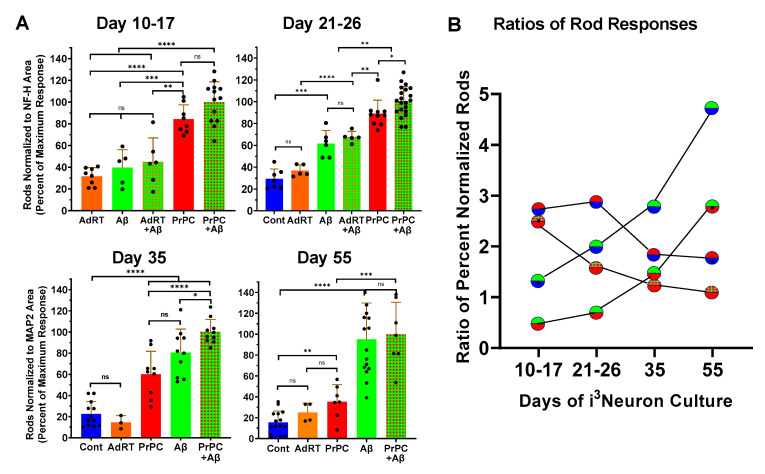
Quantification of cofilin–actin rods induced by PrP^C^ expression ± Aβd/t in i^3^Neurons at different ages. (**A**) I^3^Neurons were infected with 30 moi of AdPrP^C^ or with a control adenovirus (AdRT) at least 3 days prior to overnight treatment with Aβd/t. The rod response was quantified in each stitched image and normalized to the area of either NF-H or MAP2 immunofluorescence as a measure of neurite density. Rod responses from individual cultures were normalized to the percentage of the maximum response within each experiment, and then data from multiple experiments (*n* = 3 except for Day 35 and Day 55 AdRT, in which *n* = 2) were combined. Each point represents an individual culture in which an area of 49 fields (20× objective) were stitched for rod quantification. No Aβd/t-induced rod response was observed in cultures 17 days or younger. In cultures between 21 and 26 days, Aβd/t gave a rod response about twice that of controls (spontaneous rods), and by 35 days, the response was 3–4 times that of the control values, although when combined with PrP^C^ expression, the response was still significantly greater. By Day 55, the rod response to Aβd/t was completely independent of ectopic PrP^C^ expression. * *p* < 0.05; ** *p* < 0.01; *** *p* < 0.001; **** *p* < 0.0001; ns not significantly different. (**B**) The ratios of the average rod responses at different ages are plotted in the upper and lower semi-circles using the same colors for each treatment as in the bar graphs. The rod response of Aβd/t/control (green/blue) shows the largest increase over time, with a smaller increase for Aβd/t/PrP^C^ (green/red). Although the ratio of PrP^C^-induced rods to control (red/blue) declines somewhat, it is not as substantial as the decline in the ratio of (Aβd/t + PrP^C^)/PrP^C^ (red-green hatched/red dots), which drops from 2.5 to nearly 1, demonstrating that by Day 55, ectopic PrP^C^ no longer contributes to Aβd/t-induced rods.

**Figure 7 biomedicines-11-02942-f007:**
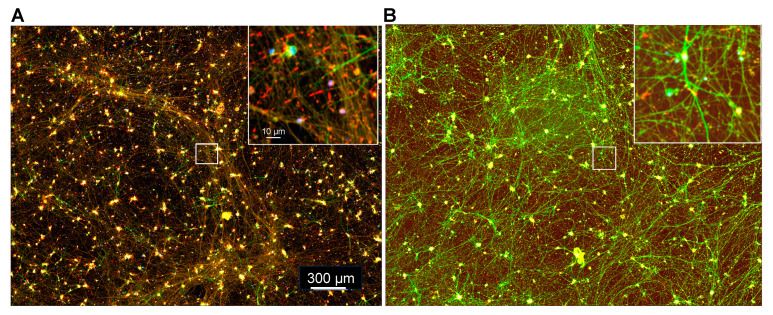
Visualization of cofilactin rods in Day 55 i^3^Neurons. (**A**) i^3^Neurons were infected on Day 50 with 30 moi of AdPrP^C^ and treated on Day 54 with Aβd/t. After fixation, methanol permeabilization, and immunolabeling for MAP2 (green) and cofilin (red), along with DAPI for nuclear staining, the cells were imaged with a 20× objective to obtain a 7 × 7 array that was stitched. The inset shows the enlargement of the boxed region with many cofilin-immunolabeled rods. Most nuclei (blue) remain scattered, with some small clumps of soma staining bright yellow. Rods scored in this image totaled 1961, well above the average of 1150 rods per array for replicates in this experiment. (**B**) Uninfected Day 55 i^3^Neurons processed identically to the image above. Rods scored in this image totaled 139 (about 12% of the average maximum), within the range usually observed for spontaneous versus induced rods at Day 55. Although the intensity of MAP2 immunolabel is weaker in the rod-containing culture (**A**), if the threshold for MAP2 is adjusted to enhance the weaker staining using color-separated images, the percentage of labeled area is nearly identical, as is better observed within the two enlarged regions (insets).

**Figure 8 biomedicines-11-02942-f008:**
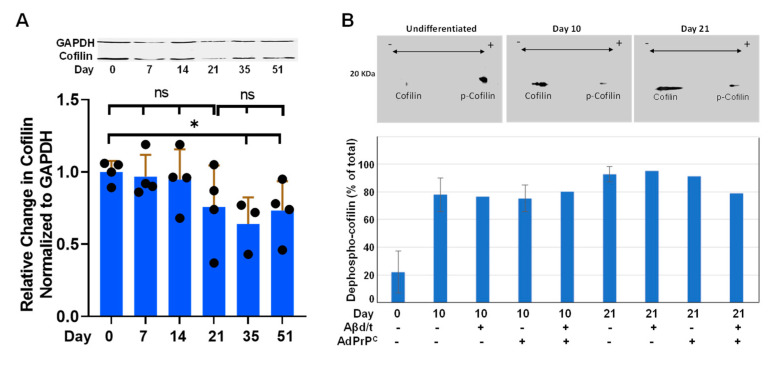
Changes in total and active cofilin during i^3^Neuron development. (**A**) Sample immunoblot and quantification from several blots (*n* = number of dots on bars) normalized to a cofilin/GAPDH ratio of 1.0 at Day 0. A slight decline in total cofilin is statistically significant (*p* < 0.05) on Days 35 and 51, but the values at Day 21 are not significantly different from either younger or older cultures, suggesting that the rod response in Figure 6 is not due to changes in total cofilin. * *p* < 0.05; ns not significantly different. (**B**) Sample 2D immunoblots showing the dramatic shift from ~80% of the non actin-binding phosphorylated-ser3 cofilin (*p*-cofilin) before outgrowth (Day 0) to ~20% *p*-cofilin by Day 10. One antibody (rabbit 1439) recognizing an epitope distant from the phosphorylation site visualizes both phosphorylated and unphosphorylated forms with equal affinity [23,41]. The major shift to the active actin-binding form of cofilin occurs long before i^3^Neurons form rods in response to Aβd/t. Error bars (sd) are from triplicate samples (*n* = 3). Where no bar is shown, the result is from a single 2D immunoblot. Samples treated Aβd/t or infected with adenovirus for PrP^C^ expression are indicated by a + under the bar. Compared to untreated cultures, neither expression of PrP^C^ (cells infected with AdPrP^C^ at 30 moi 4 days before harvest) nor treatment with Aβd/t (~1 nM) one day before harvest, or both treatments combined, induced an appreciable change in p-ser3 cofilin at Days 10 or 21.

**Figure 9 biomedicines-11-02942-f009:**
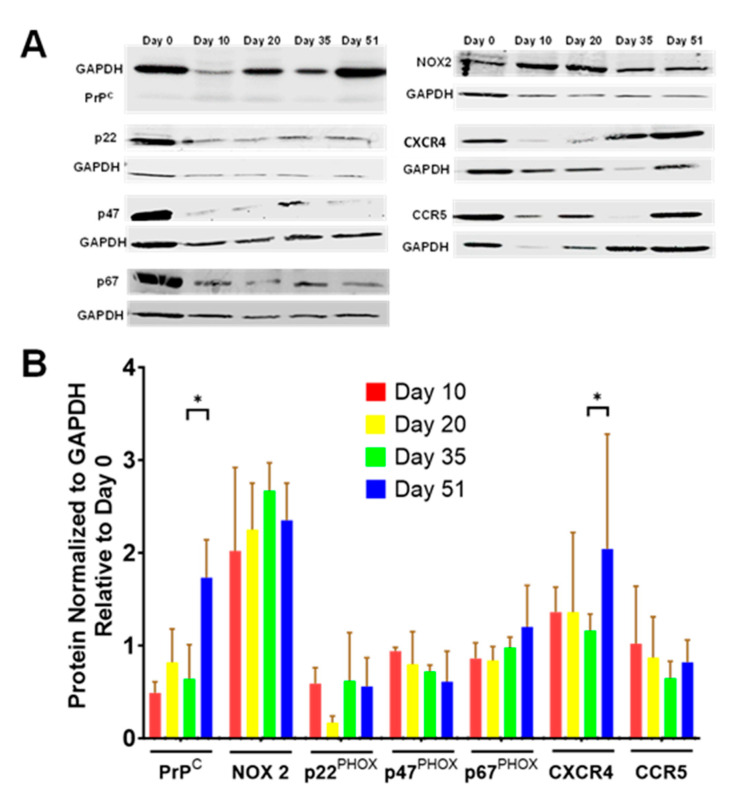
Expression of proteins in the PrP^C^/NOX-dependent rod signaling pathway during i^3^Neuron development. (**A**) Example of immunoblots for specific proteins and GAPDH loading control. (**B**) Quantified protein expression data come from immunoblots of extracts from triplicate cultures (*n* = 3). An aliquot of one sample from each harvesting day was used to make the example summary gels in (A). Band intensities were normalized to GAPDH and are expressed relative to amounts at Day 0. The CXCR4 blots are performed with an antibody against the 47 and 49 kDa subunits, and the CCR5 blots are performed with an antibody against the 40 kDa subunit. * *p* < 0.05.

**Figure 10 biomedicines-11-02942-f010:**
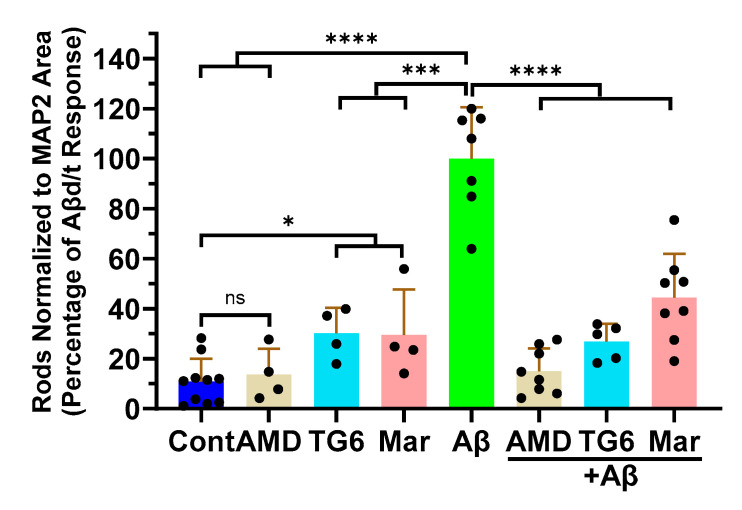
Effects of chemokine/cytokine antagonists AMD3100 (AMD) and Maraviroc (Mar) and the NOX inhibitor TG6-227 (TG6) on Aβd/t-induced rod formation in Day 55 i^3^Neurons. For responses to AMD (33 nM) or TG6 (1 µM), Aβd/t was added to cultures for 5 days, with the antagonists added for the final 24 h. For Mar (33 nM), Aβd/t was added for 24 h with the drug. In the absence of Aβd/t, the antagonists had little effect on rod formation compared to the untreated cultures. The rod response to Aβd/t was significantly above the control, AMD (both *p* < 0.0001), TG6, and Mar (both *p* < 0.001) treatments. All antagonists significantly (*p* < 0.0001) reduced the Aβd/t-induced rod response to levels of the antagonist alone. Rod numbers normalized to the MAP2-immunolabeled area were combined from 3 experiments (*n* = 3) over 18 months by adjusting each group to the average of the maximum rod response in their experimental group. ns, not significantly different, * *p* < 0.05, *** *p* < 0.001, **** *p* < 0.0001 compared between treatments as shown. There are no significant differences between each drug treatment alone (bars left of Aβ) and their counterparts in the presence of Aβd/t.

**Figure 11 biomedicines-11-02942-f011:**
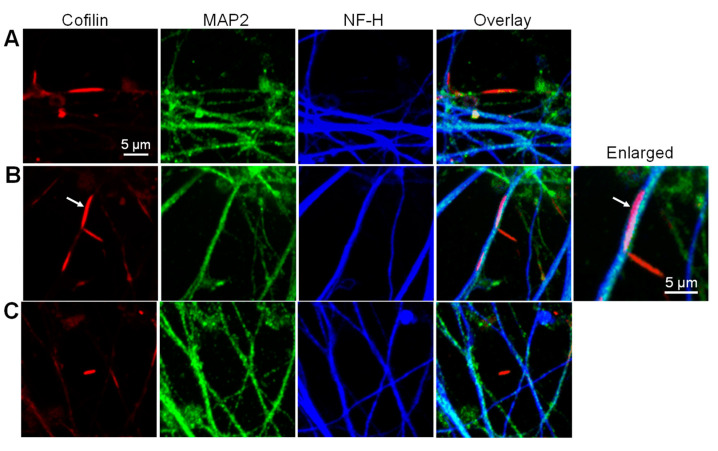
Localization of rods to dendrites or axons. Confocal images of rods immunolabeled for cofilin in Day 55 i^3^Neurons that were also immunolabeled for MAP2 (dendrites) and NF-H (axons). (**A**) Cofilactin rod in dendrite, in which MAP2 staining near the rod is disrupted. (**B**) Cofilactin rod in axon of a multi-neurite bundle. One rod (arrow) appears to be in an axon that wraps around a dendrite and another axon but requires scrolling through the image stack to see this relationship. (**C**) Rod that appears to be outside of any MAP2- or NF-H-immunolabeled process, presumably because the cytoskeletal elements within it were severely disrupted prior to fixation.

**Figure 12 biomedicines-11-02942-f012:**
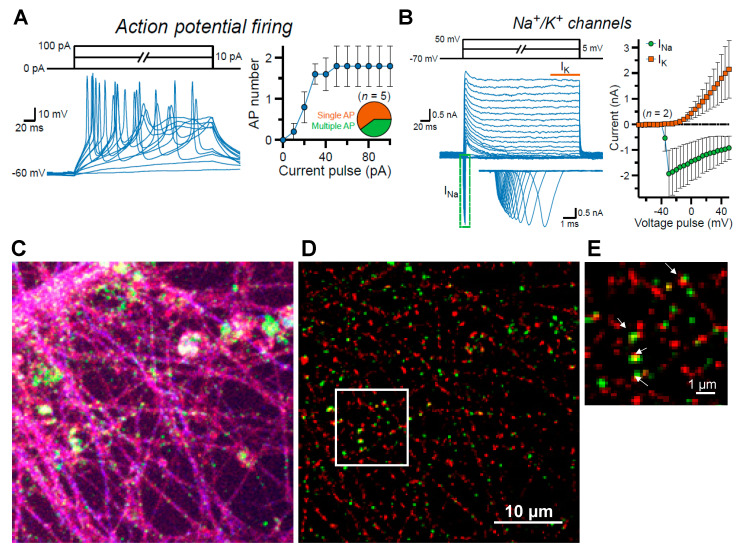
Glia-free i^3^Neurons exhibit intrinsic neuronal properties and synaptic morphologies. Day 48 i^3^Neurons, cultured in a glia-free environment with glia-conditioned medium, were characterized by electrophysiology. (**A**) Example recordings (left) in current-clamp mode, and the total number of action potentials (APs) plotted as a function of injected current amplitude (right); the pie chart indicates the fraction of cells with single (orange) or multiple (green) APs. (**B**). Representative traces (left) of Day 48 i^3^Neurons in voltage-clamp mode and the average amplitudes (right) of either Na^+^ or K^+^ channel currents (I_Na_ or I_K_, respectively); the inset (left, boxed region in green) is further magnified below to better visualize inward I_Na_ peaks, whereas I_K_ values were averaged for 50 ms (orange line). Summary graphs are plotted as means ± SEM and include the number of neurons patched per experiment. Results indicate a robust expression of voltage-gated ion channels in differentiated neurons by 48 days in a glia-free culture. (**C**) Projection image of a confocal stack (100× objective) of fixed Day 55 i^3^Neurons immunolabeled for VGLUT1 (red), PSD95 (green), and MAP2 (purple). PSD95 puncta are in dendritic spines that overlay dendrites. (**D**) Deconvolved image in (**C**) without the MAP2 immunolabeling to focus on axonal and dendritic regions, where over 19% of dendritic post-synaptic PSD95 puncta (green) abut or overlap with the pre-synaptic marker VGLUT. (**E**) Enlarged boxed region from (**D**) with arrows pointing to regions where VGLUT and PSD95 abut or overlap.

## Data Availability

All data, including access to hundreds of thousands of original fluorescent images acquired during our investigations described herein, can be obtained by contacting the corresponding author.

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
