# Peer review of "Characterization of a Human Neuronal Culture System for the Study of Cofilin–Actin Rod Pathology"

_biomedicines, 2023, doi:10.3390/biomedicines11112942_

Round 1
Reviewer 1 Report
Comments and Suggestions for Authors
In this manuscript (ID biomedicines-2670601), entitled “Characterization of a Human Neuronal Culture System for Study of Cofilin-actin Rod Pathology”, authors Tahtamouni et al studied cofilactin rod pathology in human neuronal cultures. They optimized the culture methods for the neuronal iPSC techniques. The results are interesting. However, there are several major concerns, which are listed in the following paragraphs:
1. The background information regarding WTC-11 cell-line should be provided. The authors mentioned “i3Neurons are an ideal model in which to study the function of rods”. What is the advantage and limitation of this model as compared with other models mentioned in the Introduction?
2. Regarding the data analysis, the sample size should be provided clearly in the figure legend, such as Fig 6, Fig 8, Fig 9 (n=?).
3. In Fig 11C, I could not see MAP2-green in those neurons. The VGLUT (red) and PSD95 (green) are rarely overlap. Those results don’t support the conclusion.
4. The electrophysiology was recorded from i3Nueons cultured without glia. How about the neurons cultured with glia? This data is important to support the results from other observation in this study.
Comments on the Quality of English LanguageThe manuscript should be edited carefully in grammar and spelling.
Author Response
Response to Reviewer 1
We thank the reviewer for finding the results of this study to be of interest. We offer the following responses to each of this reviewer’s concerns:
- The background information regarding WTC-11 cell-line should be provided. The authors mentioned “i3Neurons are an ideal model in which to study the function of rods”. What is the advantage and limitation of this model as compared with other models mentioned in the Introduction?
We provided references to the published information concerning the development of the WTC-11 cell line and mentioned its advantage over the human embryonic cell lines that needed to be virally infected to express the glutamatergic transcription factor Ngn2. With regard to other models mentioned in the introduction (SH-SY5Y cells and human ES cells), these are covered in the Supplementary Materials showing neither of these lines developed cofilin-actin rods in response to Aβd/t, although we did not follow the hES cell-derived neurons beyond 28 days, by which time the i3Neurons had started to show a rod response. The quote from the reviewer’s comment is in our final paragraph of the discussion and reflects our opinion of the results of our manuscript, describing the only human cell model to date in which rods induced through the PrPC/NOX pathway are shown to occur.
- Regarding the data analysis, the sample size should be provided clearly in the figure legend, such as Fig 6, Fig 8, Fig 9 (n=?).
The n values have been added in Figure 6 and Figure 8. Old Figure 9 is now Figure 10 and the n value is also provided.
- In Fig 11C, I could not see MAP2-green in those neurons. The VGLUT (red) and PSD95 (green) are rarely overlap. Those results don’t support the conclusion.
There was no green MAP2 in the image. It was labeled as blue but actually appeared purple in the overlayed image and so the color given in the figure legend was changed from blue to purple. We thank the reviewer for pointing out the difficulty in seeing the contacts between PSD95 and VGLUT. To help support our conclusions, we added quantitative information concerning the contact between PSD95 and VGLUT puncta (19.2%) and provided additional information (Table S1) and added a radial cross-correlation analysis (Figure S6) that strengthens our argument that these are specific (non-random) associations.
- The electrophysiology was recorded from i3Nueons cultured without glia. How about the neurons cultured with glia? This data is important to support the results from other observation in this study.
Please note that the morphological and functional properties of i3Neurons co-cultured with glia have already been extensively characterized by previous studies (e.g., Fernandopulle et al., Curr. Protoc. Cell Biol. 2018; Burlingham et at., Nat. Commun., 2022). In this current manuscript, it was particularly important to assess whether these cells can acquire and retain neuronal identities (e.g., action-potential firing, voltage-gated Na+/K+ channel expression) when maintained for prolonged period of time in the absence of astrocytes. Since our assays for rod response in i3Neurons comes from glia-free cultures (mainly to monitor neuronal phenotype, and avoid background cofilin immunolabeling from glia), our electrophysiological assessments provided crucial information that these neuronal cultures were functionally mature, and thus, justified the use of this system.
Reviewer 1 also suggested that the manuscript should be edited carefully in grammar and spelling. However, no specifics were provided. Furthermore reviewer 2 stated: “Overall the manuscript is excellently written and understandable.” Thus, all authors have carefully read the manuscript and several changes in the text were proposed, many of which have been incorporated for clarity. Although we found very few examples of either spelling or grammatical corrections, we are not infallible in this regard and will accept corrections if found by the editorial staff.
Reviewer 2 Report
Comments and Suggestions for Authors
In this study, Tahtamouni et al. established an in-vitro cell culture system to study human neuronal cells and their responses during Cofilin-Acton Rod pathology. The authors provide a useful cell tool and observed interesting findings concerning the interplay between the Nox2/PrPc system in their culture system. They used a broad spectrum of techniques to solidify their findings. From the experimental point of view only one minor change should be made to complete the already compelling story (see minor experimental points).
Overall the manuscript is excellently written and understandable. The methods are accurately described and the findings are clearly depicted and discussed. I have no further additions concerning the text, the format or the spelling.
Minor experimental points:
· Since the authors showed the original blots in Supplementary Figure S6, this Figure should be placed above the Table 1 in the main text. Blots are even more important for data consolidation than the quantifications. Instead of the Table 1 the authors could also depict the quantification values as bar charts. This would make the statement much clearer.
Author Response
We thank reviewer 2 for their analysis of the contribution of this manuscript to human cell cultures for understanding cofilin-actin rod pathology. This reviewer made one suggestion:
Since the authors showed the original blots in Supplementary Figure S6, this Figure should be placed above the Table 1 in the main text. Blots are even more important for data consolidation than the quantifications. Instead of the Table 1 the authors could also depict the quantification values as bar charts. This would make the statement much clearer.
As suggested by this reviewer, we have eliminated Table 1 and moved the immunoblots from Figure S6 into the main text (new Figure 9A) along with creating a bar graph (new Figure 9B) showing changes in expression of each of the proteins examined between days 10 and 51 of culture.
Round 2
Reviewer 1 Report
Comments and Suggestions for Authors
No further recommendation